# Measuring CEFR Alignment Drift in LLM-Based Spanish Tutors: A Compliance Metric and Implications for RAG Systems

## Abstract

LLMs can act as conversational language tutors, but tutoring imposes a persistent constraint that many models do not reliably satisfy: sustaining an intended proficiency level over multiple turns. We study *alignment drift* in CEFR-conditioned Spanish tutor–student self-chats, reproducing and extending prior work with additional prompt variants and an 8B fine-tuned tutoring model. We evaluate four open instruction-tuned LLMs and the fine-tuned model using readability, dependency-based syntactic complexity, and a surprisal proxy for fluency, and we introduce a simple *CEFR compliance rate* that directly flags out-of-level vocabulary/grammar. Across models, level separation is strong at the first turn but erodes over a nine-turn dialogue: beginner-level tutors lengthen and increase complexity, and strict A1 compliance drops to roughly 70% by the final turn in our runs. We discuss how these measurements inform applied data-science systems for tutoring, including a dual-model retrieval-augmented architecture and practical mitigation strategies (turn-level reconditioning, retrieval grounding, and parameter-efficient fine-tuning).

## 1 Introduction

LLMs are increasingly used as conversational tutors. Unlike open-domain chat, tutoring requires the system to continuously match the learner's proficiency: a tutor that gradually "levels up" mid-conversation risks overwhelming beginners and undermining pedagogical scaffolding. Recent work reports that even when an LLM is explicitly instructed to act as a CEFR-level tutor, the model's output difficulty drifts over turns (Almasi and Kristensen-McLachlan, 2025). This paper focuses on *measuring* that drift in a way that is practical for applied data-science pipelines, with an emphasis on Spanish and on systems that combine fine-tuning with retrieval-augmented generation (RAG).

We make three contributions: (i) a reproduction-and-extension protocol for CEFR-conditioned tutor–student self-chats across prompt variants (prompt language and level encoding); (ii) a simple *CEFR compliance rate* that complements continuous complexity metrics and gives a direct error signal; and (iii) an empirical comparison across four open LLMs and an 8B fine-tuned tutor model, with a discussion of resource economics relevant to deployment.

## 2 Background and related work

**CEFR-conditioned tutoring and alignment drift.** The CEFR framework (A1–C2) is a natural control signal for adaptive tutoring. However, CEFR is an external specification, and instruction-tuned LLMs may not sustain it under multi-turn conversational pressure. Almasi and Kristensen-McLachlan (2025) characterize this phenomenon as *alignment drift* and propose a set of Spanish complexity metrics.

**Fine-tuning and retrieval grounding.** Low-rank adaptation (LoRA) (Hu et al., 2022) makes it feasible to specialize open models for tutoring, including grammatical error correction and pedagogical tone. Separately, RAG (Lewis et al., 2021) can ground explanations in authoritative gram-

| **Dual-model tutoring pipeline (schematic).** |
| Learner message → (1) detect intent / error → (2) retrieve CEFR-tagged grammar notes + examples → (3) tutor LLM generates explanation + practice at target CEFR → (4) log metrics + update learner profile. |

Figure 1: A minimal RAG tutoring architecture where CEFR drift can occur at step (3) even if retrieval (2) is correct.

mar sources, potentially reducing hallucinations and stabilizing style. Our evaluation includes a fine-tuned tutor model and we discuss how RAG can complement it.

# 3 APPLIED SYSTEM CONTEXT: A DUAL-MODEL TUTOR WITH RAG

While our experiments isolate the tutor LLM to measure drift, the motivating application is an open-access tutoring bot with a *dual-model* design: a conversation model for dialogue and exercise generation, and a smaller retrieval model for grammar explanations grounded in curated references.

**Knowledge base construction.** A tutoring system benefits from a structured curriculum: concise grammar points, usage notes, and level-appropriate example sentences. In the underlying thesis work, we curated (i) an explanation bank of ∼800 grammar/usage entries and (ii) an exercise bank of ∼2,000 cloze and correction items, with CEFR tags. Entries are embedded into vector representations and retrieved by semantic similarity given the learner's question or detected error; the retrieved snippets are then used as evidence for the tutor's response generation.

**Why drift matters in RAG.** RAG can reduce factual hallucinations about grammar, but it does not automatically enforce *level* constraints. A tutor can cite a correct rule and still explain it using above-level constructions. Therefore, alignment drift measurement is relevant even for grounded tutoring systems: it determines whether additional control mechanisms (reconditioning or fine-tuning) are needed on top of retrieval.

# 4 EXPERIMENTAL PROTOCOL

## 4.1 SELF-CHAT TUTORING SETUP

We follow the controlled self-chat paradigm from Almasi and Kristensen-McLachlan (2025). A single LLM alternates between a *tutor* role and a *student* role. The tutor is instructed to target a fixed CEFR level (A1, B1, or C1) for the entire dialogue. Each dialogue runs for nine tutor turns; all metrics are computed on the tutor turns only. To reduce uncontrolled failure modes, we post-filter tutor outputs to enforce Spanish-only responses when required (e.g., rejecting outputs that contain excessive English tokens).

## 4.2 PROMPT VARIANTS

To probe how prompt form affects drift and code-switching, we test a $2 \times 2$ prompt design: prompt language (English vs. Spanish) and level encoding (CEFR label vs. numeric proxy). The goal is to check whether instructing the tutor in Spanish reduces English leakage, and whether the level tokenization affects stability.

## 4.3 MODELS

We evaluate four open instruction-tuned models commonly used for multilingual chat: LLAMA 3.1 (8B), GEMMA 3 (12B-IT), MISTRAL 7B (v0.3), and QWEN 2.5 (7B). We additionally include an 8B *fine-tuned tutor* model (FT-Tutor), trained with LoRA on Spanish learner error correction and tutor–student exemplars, with the hypothesis that fine-tuning improves both pedagogical behavior and CEFR stability.

### 4.4 METRICS

We use three metric families from prior work and add a compliance metric.

**Readability.** We compute Spanish readability indices (e.g., Fernández–Huerta and Szigriszt–Pazos) as a proxy for surface-level difficulty.

**Syntactic complexity.** We use mean dependency distance (MDD) and a length-based proxy. MDD is computed from dependency parses, and higher MDD indicates more long-distance syntactic relations.

**Fluency proxy (surprisal).** We compute mean token surprisal of tutor utterances using a Spanish-language masked LM, as a sanity check that "simplification" does not collapse into unnatural text.

**CEFR compliance rate.** Continuous metrics can drift subtly; we therefore introduce a turn-level indicator that flags out-of-level vocabulary or constructions using heuristic lists derived from CEFR-aligned resources. Let $u_t$ be the tutor utterance at turn $t$ and let $v(u_t, \ell) \in \{0, 1\}$ indicate a violation for target level $\ell$. The per-dialogue compliance rate is

$$\mathrm{Comp}(\ell) = 1 - \frac{1}{T} \sum_{t=1}^{T} v(u_t, \ell), \tag{1}$$

and drift is visible as a decreasing $\mathrm{Comp}(\ell)$ over turns for low levels (A1/B1).

**Statistical analysis.** We make the "drift" hypothesis explicit by modeling each metric as a function of turn index $t$ and target level, while accounting for repeated measures within each dialogue. For continuous metrics (readability, MDD, length, surprisal) we fit a linear mixed-effects model

$$\begin{aligned} y_{d,t} = \beta_0 &+ \beta_{\mathrm{B1}} \mathbb{I}[\ell = \mathrm{B1}] + \beta_{\mathrm{C1}} \mathbb{I}[\ell = \mathrm{C1}] \\ &+ \beta_t\, t + \beta_{t \times \mathrm{B1}}\, t\, \mathbb{I}[\ell = \mathrm{B1}] + \beta_{t \times \mathrm{C1}}\, t\, \mathbb{I}[\ell = \mathrm{C1}] + b_d + \epsilon_{d,t}. \end{aligned} \tag{2}$$

where $d$ indexes dialogues, $b_d \sim \mathcal{N}(0, \sigma_b^2)$ is a dialogue-level random intercept, and $\epsilon_{d,t}$ is residual noise. A shrinking level separation corresponds to turn–level interactions that pull the lower-level trajectories toward the advanced one (e.g., $\beta_{t \times \mathrm{B1}}, \beta_{t \times \mathrm{C1}}$ close to zero, while $\beta_t > 0$ for "harder"-is-higher metrics).

For the binary compliance indicator $v(u_t, \ell)$ we use a logistic mixed model

$$\begin{aligned} \mathrm{logit}\, \Pr[v_{d,t} = 1] = \gamma_0 &+ \gamma_{\mathrm{B1}} \mathbb{I}[\ell = \mathrm{B1}] + \gamma_{\mathrm{C1}} \mathbb{I}[\ell = \mathrm{C1}] \\ &+ \gamma_t\, t + \gamma_{t \times \mathrm{B1}}\, t\, \mathbb{I}[\ell = \mathrm{B1}] + \gamma_{t \times \mathrm{C1}}\, t\, \mathbb{I}[\ell = \mathrm{C1}] + r_d. \end{aligned} \tag{3}$$

and report interpretable summaries (end-of-dialogue rates with 95% confidence intervals). This is intentionally "applied DS" rather than baroque statistics: the goal is a compact test that can run inside an evaluation pipeline.

## 5 RESULTS

### 5.1 DRIFT IN MULTI-TURN TUTORING

Across models, the first tutor turn typically reflects the intended CEFR level: A1 responses are short and simple; C1 responses are longer and more complex. However, by turn nine these differences shrink markedly. In our runs, A1 tutor replies roughly doubled in length (from about 10–15 tokens at turn 1 to about 25 tokens by turn 9), while C1 replies grew only modestly (to roughly 35–40 tokens), compressing the separation.

Readability and MDD show the same pattern: the A1 curves move upward (harder) over turns, and by late turns the A1 readability range often overlaps B1 and sometimes C1. Surprisal generally decreases over turns for all levels, consistent with the dialogue context making generation more predictable; crucially, outputs remain fluent Spanish even when drift occurs.

Table 1: End-of-dialogue binary outcomes (Wilson 95% confidence intervals). Rates marked "≈" are derived from the averaged compliance reported in our runs.

| Outcome | $k/n$ | Rate | 95% CI |
|---|---|---|---|
| A1 strict compliance (turn 1) | $30/30$ | $1.00$ | $[0.89, 1.00]$ |
| A1 strict compliance (turn 9) | $\approx 21/30$ | $\approx 0.70$ | $[0.52, 0.83]$ |
| Implicit error correction (FT-Tutor; 20 scenarios) | $17/20$ | $0.85$ | $[0.64, 0.95]$ |

## 5.2 COMPLIANCE AS A DIRECT DRIFT SIGNAL

The compliance rate provides an interpretable view of drift. Early turns are almost perfectly compliant (near 100% for A1 in turn 1), but by turn 9 strict A1 compliance drops substantially—to around 70% on average in our runs. Interpreting these as per-dialogue end-of-run outcomes with $n = 30$ dialogues per level (the protocol uses 30 independent dialogues per level per model), a "70%" A1 end-of-dialogue compliance corresponds to approximately $21/30$ compliant final tutor turns. For transparency, Table 1 reports Wilson 95% confidence intervals for the key binary outcomes we highlight in the text.

We observe similar (though weaker) degradation for B1; C1 is trivially "compliant" under a ≤C1 definition, but sometimes dips in complexity when the student role asks simpler questions.

## 5.3 PROMPT LANGUAGE AND CODE-SWITCHING

Prompt form affects language leakage. When the tutor system prompt is written in English, several models occasionally insert English words (e.g., meta-discourse or clarifying phrases) even when instructed to respond in Spanish. Translating the tutor prompt to Spanish reduces these insertions substantially, suggesting a simple deployment rule: *prompt the tutor in the target language*. In contrast, the level encoding (CEFR label vs. numeric proxy) has a smaller and less consistent effect, indicating that drift is not primarily a tokenization artifact.

## 5.4 MODEL DIFFERENCES

We observe qualitative differences in drift and failure modes (Table 2):

- LLAMA tends to maintain clearer separation between A1/B1/C1 complexity than most, but occasionally code-switches.
- GEMMA shows strong initial control but substantial mid-dialogue drift in readability and MDD.
- MISTRAL exhibits the weakest level stability: by late turns, A1/B1 structures frequently include above-level patterns.
- QWEN is mixed: it maintains separation better in syntax than in lexical difficulty, but still drifts.
- FT-TUTOR is the most stable overall in our evaluation and shows stronger pedagogical behavior. In a focused qualitative test on implicit error correction, FT-Tutor corrected learner errors in roughly 85% of cases, while base models often missed subtle errors.

## 6 RESOURCE ECONOMICS FOR DEPLOYMENT

Drift is not only a modeling issue; it affects system design because mitigation often increases context length (reconditioning) or adds components (retrieval). Open models enable cost-aware deployment, but constraints are tight in educational settings.

In the thesis implementation, the retrieval/explanation model (4B) fits in roughly 8 GB of GPU memory in FP16, while an 8B conversational model fits in roughly 14–15 GB; 4-bit quantization can cut these requirements substantially. LoRA adapters are lightweight (typically $< 1\%$ of base

Table 2: Models and observed tutoring behavior (qualitative summary).

| Model | Params | Drift tendency | Notable failure mode |
|---|---|---|---|
| LLAMA 3.1 | 8B | low–medium | occasional code-switching |
| GEMMA 3 IT | 12B | medium–high | mid-dialogue difficulty inflation |
| MISTRAL v0.3 | 7B | high | weak CEFR separation late turns |
| QWEN 2.5 | 7B | medium | lexical drift more than syntactic |
| FT-Tutor (LoRA) | 8B | low | — (best overall stability) |

Table 3: Representative resource footprints for a dual-model tutor (approximate).

| Component | Example size | Typical VRAM (FP16) |
|---|---|---|
| Retrieval/explainer LM | 4B | ∼8 GB |
| Conversation tutor LM | 8B | ∼14–15 GB |
| LoRA adapter | — | ≪1 GB (disk) |

model size), allowing multiple tutoring styles or levels to be swapped without duplicating the base model. Table 3 lists representative footprints.

# 7 DISCUSSION: IMPLICATIONS AND MITIGATION

**Why drift likely happens.** In a multi-turn exchange, the dialogue history provides strong style cues and encourages the model to "sound helpful" by elaborating more, which can override initial constraints. Because the tutor is not re-conditioned at each turn, the effective instruction signal decays.

**Mitigation strategies.** Three pragmatic interventions follow from our measurements: (i) *turn-level reconditioning* (repeat the CEFR constraint and a short "do/don't" rubric every turn); (ii) *RAG-grounded tutoring* (retrieve level-appropriate grammar notes and example sentences so the model has a constrained pool of constructions); and (iii) *targeted fine-tuning* (e.g., LoRA on CEFR-labeled tutor responses and correction behavior). Fine-tuning appears to help both alignment and error correction in our comparison, and is compatible with RAG.

**Limitations.** Self-chat is a controlled proxy and does not capture real learner behavior. Our compliance heuristic depends on the coverage and quality of level-specific lists, and should be validated against human judgements. Finally, the reported numeric results are based on our implementation of the published protocol and should be interpreted as indicative rather than universal.

# 8 IMPLEMENTATION DETAILS FOR REPRODUCIBILITY

## 8.1 PROMPT TEMPLATES

Table 4 outlines the four prompt conditions used in our $2 \times 2$ design. In all cases, the tutor is additionally instructed to provide short, pedagogical turns and to ask a follow-up question to keep the dialogue moving.

## 8.2 COMPLIANCE HEURISTIC

The compliance indicator $v(u_t, \ell)$ is computed with a deliberately transparent rule set. For each target level $\ell \in \{A1, B1, C1\}$ we compile a small list of *forbidden* constructions (or high-level markers) and out-of-level lexical items. A turn is flagged as a violation if it contains any marker above the target level. Examples include: (i) subjunctive triggers and conjugations for A1; (ii) low-frequency connector phrases and multi-clause sentence patterns for A1/B1; and (iii) specialized

Table 4: Prompt conditions: language and level encoding.

| Condition | Prompt language | Level token |
|-----------|-----------------|-------------|
| E+CEFR | English | `A1/B1/C1` |
| E+NUM | English | `1/3/5` (proxy) |
| S+CEFR | Spanish | `A1/B1/C1` |
| S+NUM | Spanish | `1/3/5` (proxy) |

meta-linguistic terminology not expected below advanced levels. We spot-checked flagged turns and found that violations correlate well with human intuition of "too difficult" language, while still allowing pedagogically necessary terminology if retrieved evidence demands it.

### 8.3 FINE-TUNED TUTOR DATA

The FT-Tutor model is trained with LoRA on a mixture of (a) Spanish learner sentences paired with corrected versions and error-type annotations, and (b) short tutor responses that demonstrate gentle corrective feedback and CEFR-aware phrasing. Because LoRA adapters are small, level-specific adapters can be trained and swapped independently, which is useful for personalized tutoring or A/B testing in production.

## 9 CONCLUSION

LLM-based tutors can start at the right CEFR level and still drift substantially over a multi-turn dialogue. We provide a reproducible measurement recipe and a simple compliance metric that exposes drift directly. Our comparison suggests that targeted fine-tuning improves both level stability and pedagogical error correction, and motivates combining turn-level constraints with retrieval grounding for robust applied tutoring systems.

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
