# OpenReview forum: "MEASURING CEFR ALIGNMENT DRIFT IN LLMBASED SPANISH TUTORS: A COMPLIANCE METRIC AND IMPLICATIONS FOR RAG SYSTEMS"
_mathai.club/MathAI/2026/Conference — MathAI 2026 Conference Submission_

### Official Review · Reviewer_3pcz · 2026-03-11
**Interesting educational NLP question severely undermined by insufficient validation and statistical rigor**

**Rating:** 3
**Confidence:** 4

**Review:**

Summary
This paper investigates whether LLM-based Spanish tutors maintain CEFR-level compliance across multi-turn dialogues. Using a self-chat evaluation protocol, the authors measure "alignment drift"—the tendency for tutors to produce increasingly complex language over conversation turns, particularly at lower CEFR levels (A1). They report that A1 compliance drops to ~70% by the final turn.
Strengths

Addresses a practically important question for educational NLP: do LLM tutors stay at the target difficulty level?
Multi-model comparison (Llama, Gemma, Qwen, fine-tuned model) provides breadth.
The compliance metric, while simple, is an intuitive first attempt at quantifying CEFR drift.

Weaknesses
Severely Insufficient References

Only 3 citations for a full paper. There are no references to: the CEFR framework itself, Spanish readability literature (Fernández-Huerta and Szigriszt-Pazos are mentioned but never cited), syntactic complexity theory, language assessment methodology, or prompt engineering research. This is unacceptable for any venue.

Unvalidated Core Metric

The compliance heuristic—the paper's central measurement tool—was only "spot-checked" against "human intuition." No inter-annotator agreement study, no confusion matrix against expert ratings, no validation against established CEFR assessment tools. For a paper whose entire contribution is a measurement, the measurement is unvalidated.

Statistical Rigor

The headline finding "A1 compliance drops to roughly 70%" has a 95% CI of [0.52, 0.83]. The actual compliance could be anywhere from 52% to 83%—this range is so wide it is effectively meaningless, yet the abstract presents "70%" as a precise finding.
Mixed-effects models (Eqs. 2–3) are reported without test statistics, pp
p-values, or model diagnostics
. The dismissive statement "applied DS rather than baroque statistics" is not an acceptable substitute for statistical rigor.
FT-Tutor was tested on only n=20n=20
n=20 scenarios while base models used n=30n=30
n=30. No explanation provided, yet FT-Tutor is labeled "best overall stability."


Methodological Concerns

Self-chat (LLM playing both student and tutor) is acknowledged as non-representative of real learner interactions but used as the sole evaluation method.
The binary compliance indicator (Eq. 1) treats all violations equally—a single subjunctive verb and a paragraph of C1-level text are counted the same.
Core hypothesis that dialogue history drives drift is never tested via ablation (e.g., comparing multi-turn vs. isolated single-turn responses).

Domain Issues

Subjunctive mood is listed as an A1-level violation, but expressions like "No creo que..." are standard B1-level Spanish. The heuristic's linguistic accuracy is questionable.
No comparison to human tutors as a baseline. Do human Spanish teachers maintain >90% compliance across 9 turns?

De-anonymization Risk

Line 100 references "the underlying thesis work" with specific corpus sizes (~800 grammar entries, ~2,000 cloze items). This could enable reviewer identification of authors.

Novelty

The paper explicitly states it "reproduces and extends" Almasi & Kristensen-McLachlan (2025). The extensions (additional prompt variants, one fine-tuned model) may not constitute sufficient novelty for a full paper.

Questions for Authors

Can you provide inter-annotator agreement for the compliance heuristic?
What are the pp
p-values and test statistics for the mixed-effects models?

Why was FT-Tutor tested on fewer scenarios?
Have you compared to human tutor baseline?

Overall Assessment
An interesting research question, but the core metric is unvalidated, the statistics are insufficiently reported, the confidence intervals are too wide to draw conclusions, and only 3 references are cited. Major revisions needed across all dimensions.

---

### Official Review · Reviewer_VMAc · 2026-03-12
**This paper studies CEFR alignment drift in LLM-based Spanish tutoring systems, examining whether language models maintain the intended learner proficiency level throughout a multi-turn dialogue. The authors evaluate several open instruction-tuned models and a fine-tuned tutoring model using linguistic complexity metrics and introduce a CEFR compliance rate to quantify whether generated responses remain within the target proficiency level. The study shows that although models start with good level separation, linguistic complexity tends to increase over conversation turns, causing noticeable drift from beginner-level constraints.**

**Rating:** 6
**Confidence:** 3

**Review:**

Overall, the paper provides a useful empirical observation about alignment drift in tutoring scenarios, and it proposes several mitigation strategies, including retrieval grounding, turn-level reconditioning, and parameter-efficient fine-tuning. However, while the study is practically motivated and the results are interesting, the experimental scope remains somewhat limited, and some methodological aspects, such as evaluation validity and dialogue realism could be further strengthened.


Strengths:

1. Focus on a practical and underexplored problem: The paper highlights a subtle but important challenge in educational AI systems: maintaining consistent pedagogical difficulty across multiple turns of interaction. This problem is often overlooked in general LLM evaluation, making the study both timely and practically relevant.
2. Clear operationalization of CEFR alignment: Introducing a CEFR compliance rate as an evaluation metric is a useful contribution. It provides a concrete way to quantify whether generated language exceeds the target proficiency level, which could be valuable for future tutoring system evaluations.
3. Multi-metric linguistic analysis: The evaluation combines readability measures, syntactic dependency complexity, and fluency proxies, giving a reasonably comprehensive view of linguistic difficulty rather than relying on a single metric.
4. Practical system implications: The discussion on mitigation strategies, such as retrieval-augmented grounding and turn-level reconditioning, helps connect the empirical findings to real system design considerations.


Weaknesses:

1. Limited experimental realism: The evaluation is primarily conducted using self-chat tutor–student simulations, which may not accurately represent interactions with real learners. Real student prompts could introduce variability that significantly affects alignment drift.
2. Evaluation framework could be more rigorous: Although multiple linguistic metrics are used, it is not entirely clear how well these metrics capture true CEFR proficiency levels. A stronger validation such as comparison with human CEFR annotations—would increase confidence in the evaluation.
3. Scope of models and languages is narrow: The study focuses primarily on Spanish and a limited set of models. While this keeps the study manageable, the results may not generalize to other languages or tutoring contexts.
4. Mitigation strategies are discussed but not empirically tested: The paper suggests several architectural solutions (e.g., dual-model RAG pipelines), but these are not experimentally evaluated within the paper, making the recommendations somewhat speculative.


Recommendations:

1.	Include human validation of CEFR levels: Comparing automated metrics with expert CEFR annotations would strengthen the credibility of the evaluation.
2.	Test the proposed mitigation strategies empirically: Implementing and benchmarking techniques such as retrieval grounding or turn-level conditioning would make the paper more impactful.
3.	Expand evaluation to real student interactions: Using learner prompts or educational datasets would better demonstrate the practical relevance of the findings.
4.	Discuss generalization to other languages and tutoring tasks: Clarifying whether the observed drift is language-specific or model-specific would help position the work within broader AI-education research.

---

### Official Review · Reviewer_95kr · 2026-03-12
**Empirical study with incomplete statistical analysis**

**Rating:** 3
**Confidence:** 4

**Review:**

$Goal$

The authors follow the paper Almasi, M., and Kristensen-McLachlan, M. (2025). Alignment drift in CEFR [Common European Framework of Reference]-prompted LLMs for interactive Spanish tutoring arXiv:2505.08351.

The goal is to measure alignment drift and provide recommendations for a DAG architecture and mitigation strategies to reduce it.

$Contributions$

The authors summarize their contributions promptly (reformulated):

1. Reproduction and extension of the CEFR protocol.
2. Provide a simple CEFR compliance rate.
3. Provide an empirical study over four open LLMs and an 8B fine-tuned tutor model.

$Presentation$

There is an apparent gap in content on page 2 under Figure 1.

$Originality/Novelty$

These are limited, since the study closely follows one cited piece of previous work. There are two missed opportunities in this otherwise interesting study - generalize drift and investigate further the plausible hypothesis that fine-tuning improves both pedagogical behavior and CEFR stability.

$Soundness$

The statistical model generates a few questions. Although it is, in general, a fairly common ML modeling, there is an error.
Shrinking level separation: A level is omitted; the A-B1 separation at turn $t$ would be $-\beta_{B1} - \beta_{t \times B1}t $.
That leads to a different interpretation of the interaction.

The logistic model is also fairly standard. Given a repeated binary indicator, reporting predicted end-of-dialogue compliance rates with 95% confidence intervals is interpretable.

Unfortunately, the statistical analysis does not include its limitations (e.g., drift might be non-linear across turns). As a result, soundness is a suspect.

$Weaknesses$

Statistical analysis is incomplete and contains inaccuracies.

$Strengths$

Presentation is succinct and clear.

$Conclusion$

Given limited novelty and thus diminished significance, and inaccurate statistical analysis, hence the conclusion.

---

### Decision · Program_Chairs · 2026-03-20

**Decision:**

Accept (Poster)

**Comment:**

Dear Author(s),

On behalf of the Program Committee of the International Conference on Mathematics of Artificial Intelligence (MathAI 2026), we are pleased to inform you that your paper has been accepted for a poster presentation at MathAI 2026.

Your paper was evaluated through a rigorous two-stage review process involving both automated screening and expert review by members of the Program Committee. The reviewers recognized the quality and contribution of your work.

Important Note: The reviewers have recommended final revisions to your manuscript before the conference. Please ensure that all reviewer comments are carefully addressed in your camera-ready version. We trust that you will complete these revisions before the conference deadlines.

Presentation details:

    Format: Poster presentation

    Mode: You may present either in person (offline) at the conference venue in Sirius, Russia, or remotely via Zoom. Please indicate your preferred mode when confirming your participation.

    Conference dates: March 30 - April 3, 2026

    Website: https://mathai.club

Next steps:

    Please confirm your participation and presentation mode by replying to this email (mathai.club@yandex.ru) no later than March 15, 2026 18:00 Moscow time.

    If you plan to attend in person, the organizing committee will provide accommodation details separately.

    Please prepare your final camera-ready manuscript according to the formatting guidelines available at https://mathai.club and upload it to OpenReview by March 15, 2026 18:00 Moscow time. Ensure that all reviewer feedback has been incorporated into this final version.

Should you have any questions regarding the program, logistics, or your presentation, please do not hesitate to contact us.

We look forward to your contribution to MathAI 2026.

With kind regards,

MathAI 2026 Program Committee
International Conference on Mathematics of Artificial Intelligence
https://mathai.club
OpenReview: https://openreview.net/group?id=mathai.club/MathAI/2026/Conference
Telegram: https://t.me/MathAI_club
Email: mathai.club@yandex.ru